# Biomechanical Comparison of Five Fixation Techniques for Unstable Fragility Fractures of the Pelvic Ring

**DOI:** 10.3390/jcm10112326

**Published:** 2021-05-26

**Authors:** Moritz F. Lodde, J. Christoph Katthagen, Clemens O. Schopper, Ivan Zderic, Geoff Richards, Boyko Gueorguiev, Michael J. Raschke, René Hartensuer

**Affiliations:** 1AO Research Institute Davos, Clavadelerstrasse 8, 7270 Davos, Switzerland; clemens.schopper@hotmail.com (C.O.S.); ivan.zderic@aofoundation.org (I.Z.); geoff.richards@aofoundation.org (G.R.); boyko.gueorguiev@aofoundation.org (B.G.); 2Department for Trauma, Hand and Reconstructive Surgery, University Hospital Münster, Albert-Schweitzer-Campus 1, Building W1, Waldeyerstraße 1, 48149 Münster, Germany; christoph.katthagen@ukmuenster.de (J.C.K.); michael.raschke@ukmuenster.de (M.J.R.); hartensuer@uni-muenster.de (R.H.)

**Keywords:** pelvic ring fracture, external fixator, retrograde transpubic screw, S1/S2 ala–ilium screws, biomechanics

## Abstract

Background: Incidence of pelvic ring fractures has increased over the past four decades, especially after low-impact trauma—classified as fragility fractures of the pelvis (FFP). To date, there is a lack of biomechanical evidence for the superiority of one existing fixation technique over another. An FFP type IIc was simulated in 50 artificial pelvises, assigned to 5 study groups: Sacroiliac (SI) screw, SI screw plus supra-acetabular external fixator, SI screw plus plate, SI screw plus retrograde transpubic screw, or S1/S2 ala–ilium screws. The specimens were tested under progressively increasing cyclic loading. Axial stiffness and cycles to failure were analysed. Displacement at the fracture sites was evaluated, having been continuously captured via motion tracking. Results: Fixation with SI screw plus plate and SI screw plus retrograde transpubic screw led to higher stability than the other tested techniques. The S1/S2 ala–ilium screws were more stable than the SI screw or the SI screw plus external fixator. Conclusions: In cases with displaced fractures, open reduction and plate fixation provides the highest stability, whereas in cases where minimally invasive techniques are applicable, a retrograde transpubic screw or S1/S2 ala–ilium screws can be considered as successful alternative treatment options.

## 1. Introduction

Incidence of pelvic ring fractures has increased over the past four decades, as has the number of related operative treatments, particularly in the older population [1,2,3,4]. In young patients, these fractures are mainly caused by high-impact trauma, whereas in older patients they can result from low-energy falls [5,6]. The AO/OTA classification relies on the Tile classification, and distinguishes between stable, partially stable, and unstable pelvic fractures. By contrast, fractures in fragile pelvic bones without a high-impact injury mechanism—ranging between low-energy trauma and physiological stress—are defined as fragility fractures of the pelvis (FFP) [7]. The most frequently observed FFP types are type IIb—having a non-displaced sacral fracture with anterior disruption—and type IIc—representing non-displaced sacral, sacroiliac, or iliac fractures with anterior disruption [7]. Fractures of the superior pubic ramus, with simultaneous disruption of the posterior pelvic ring, result in an unstable pelvic ring. The main treatment goals in these patients are pain reduction and rapid return to mobility [5,8]. Currently, little is known about the necessity of addressing concomitant fractures of the anterior and/or posterior pelvic ring [9] in FFP type II injuries, nor which surgical technique results in the most stable solution. Different fixation methods have been described, being applied alone or in combination, including the use of a sacroiliac (SI) screw—addressing only the posterior pelvic ring—as well as an external fixator, transpubic screw, or a plate, in order to stabilise the anterior pelvic ring, with each method having its own advantages and disadvantages [8,10] (Figure 1).

To date, there is a lack of evidence for the superiority of one existing technique over another [8]. Therefore, the aim of the present study was to investigate the biomechanical competence of five fixation techniques for FFP type II pelvic injuries. It was hypothesised that plate fixation of the anterior pelvic ring would be biomechanically superior to both supplemental external and transpubic screw fixations, as well as to use of standalone S1/S2 ala–ilium screws.

## 2. Materials and Methods

An FFP type II was simulated at the right side of 50 artificial pelvises (osteoporotic bone model #LS4060, Synbone, Zizers, Switzerland), as presented in Figure 2. The anterior pelvic ring fractures lines were created via vertical osteotomies of the superior and inferior pubic ramus, located 2 cm laterally to the pubic tubercle (Figure 2).

The injury of the posterior pelvic ring was simulated by a paraforaminal transverse osteotomy of the os sacrum through the midline between the medial osseous frontier of the SI joint and the lateral osseous frontier of the first anterior sacral foramen in zone 1, according to the work of Denis et al. [11] (Figure A1). All osteotomies were set using the same saw cut template. The pelvises were assigned to 5 groups, consisting of 10 specimens each, for application of the following fixation methods (Figure 3): one SI screw (group 1, SI screw group), one SI screw plus a supra-acetabular external fixator (group 2, external fixator group), one SI screw plus a plate fixation of the superior pubic ramus (group 3, plate group), one SI screw plus one retrograde transpubic screw (group 4 transpubic screw group), and the S1/S2 ala–ilium screws (group 5, S1/S2-ala-ilum group).

Fracture reduction and fixation in the study groups was performed as follows:

In groups 1–4 predrilling for the SI screw insertion was performed with a 5.0 mm drill bit using a customised polymethylmethacrylate (PMMA, Beracryl, Suter-Kunststoffe AG, Fraubrunnen, Switzerland) aiming guide. A cannulated, fully threaded 7.5 × 90 mm titanium SI screw (Axomed GmbH, Freiburg, Germany) was inserted for fixation of the posterior ring. In addition, a 5.0 mm supra-acetabular external fixator (DePuy Synthes, Zuchwil, Switzerland,) was applied in group 2 to address the anterior fracture, using a standardised customised PMMA guide to predrill the entry points of the 5.0 mm Schanz screws with a 3.5 mm drill bit. The Schanz screws were inserted over the entire length of their thread. The anterior fracture in group 3 was addressed with a 4.5 mm 10-hole dynamic compression plate (DCP) made of stainless steel (DePuy Synthes) and precisely precontoured to the shape of the bone to ensure optimal implant fit. The plate position was standardised for each specimen (Figure A7), and 5 cortical 4.5 mm titanium screws were used for its fixation. The lateral screws were implanted supra-acetabular. The fracture of the anterior ring in group 4 was fixed with a 4.5 × 70 mm titanium retrograde transpubic screw (Axomed GmbH). Predrilling was performed with a 3.5 mm drill bit using a customised PMMA guide. In group 5—with no implementation of SI screw fixation—predrilling for the S1 pedicle screw insertion in the first sacral vertebral body was performed with a 5.0 mm drill bit, while predrilling for the S2 ala–ilium screw in the second sacral vertebral ala was performed with a 6.0 mm drill bit [12,13]. Fixation was achieved by inserting a 7.2 × 35 mm S1 pedicle screw and an 8.2 × 100 mm S2 ala–ilium screw, both made of titanium and connected with a 5.5 mm titanium rod (Silony Medical Europe GmbH, Bremen, Germany). Screw channels were set using a customised PMMA guide (Figure A2, Figure A3, Figure A4, Figure A5 and Figure A6).

In all groups, the pubic symphysis of each specimen was bonded with coarse-threaded screws for standardisation and stiffening (Figure 2) [14]. Instrumentation of all specimens was performed by the first and third authors with a radiological control after each main procedure step. Using an electronic torque screwdriver (PB 8320 A 0.4–2.5, PB Swiss Tools, Wasen/Bern, Switzerland), a standardised torque for screw tightening was set at 1.5 Nm for the SI screws, DCP screws, and S1/S2 ala–ilium screws, and at 0.5 Nm for the retrograde transpubic screw. Optical markers were glued onto the medial and lateral aspects of the fracture site of each sacrum and superior ramus, as well as at the right SI joint for motion tracking (Figure 2).

### 2.1. Biomechanical Testing

Biomechanical testing was performed on a servo-hydraulic material testing system (858 Bionix MTS Systems Corp. Eden Prairie, MN, USA) equipped with a 4 kN load cell in a setup simulating a one-legged stance position with applied load at the right hemipelvis (Figure 4) [15,16,17]. Standardisation of the hip joint loading mechanics was performed by using a unipolar hemiarthroplasty, attached to a PMMA-potted acetabular cup, the latter being press-fit fixed to the specimen. Proximally, each central body of the sacrum was fixed through a PMMA cast to an L-shaped profile featuring a radiolucent posterior section made of cotton laminates (Canevasite, HBW 2088, Amsler & Frey AG, Schinznach-Dorf, Switzerland) via two screws plus washers applied through the first row of its neuroforamina, thus allowing for anteroposterior radiographic imaging. The L-frame was connected to the load cell and the machine actuator via a hinge joint, enabling free rotation around the longitudinal anatomical axis. The specimens were aligned to the machine axis so that an axial compression force was applied through the centre of the S1 vertebral body [15]. Distally, the hemiarthroplasty was rigidly constrained to the machine base via a socket brace.

The loading protocol comprised an initial quasi-static ramp from an unloaded condition at 0 N to a 50 N preload. Subsequently, the specimens were tested until construct failure, applying progressively increasing cyclic loading with a physiological load profile of the cycles at 1 Hz. Starting from 50 N, the peak load of each cycle increased at a rate of 0.01 N/cycle, whereas the valley load was maintained at a constant level of 10 N. Test stop criterion was set to 30 mm displacement of the machine actuator.

### 2.2. Data Acquisition and Analysis

Axial stiffness was calculated from the ascending slope of the ramped load–displacement curve between 20 and 40 N. Machine data in terms of axial load and displacement were obtained from the controllers at 64 Hz. Interfragmentary displacements at the fracture sites were continuously captured via motion tracking at 50 Hz using an ARAMIS SRX camera system (ARAMIS, GOM GmbH, Braunschweig, Germany) and optical markers glued to the specimens. Interfragmentary rotations around the three principal axes–corresponding to the anatomical axes–were analysed [15]. Interfragmentary displacements at the anterior and posterior pelvic ring fracture sites (Figure A9; distance 4 and distance 6) were calculated after 500 cycles–being too big for capturing by the system cameras after higher number of cycles in groups 1 and 2. Cycles to failure, failure load and mode of failure of all specimens were evaluated with regard to the test stop criterion.

Statistical analysis among the parameters of interest was performed using IBM SPSS Statistics (v.23, IBM, Armonk, NY, USA). Normality of data distribution was checked with Shapiro–Wilk test. Univariate analysis of variance (ANOVA) with Bonferroni post-hoc tests for multiple comparisons and Kruskal–Wallis test with Bonferroni correction for multiple compari-sons were applied to detect significant differences among the study groups in case of normality and non-normality of data distribution, respectively. Level of significance was set at 0.05 for all statistical tests.

## 3. Results

Outcome measures of the investigated parameters of interest in the study groups are summarised in Table 1 and Table 2. Results from the multiple comparisons are listed in Appendix A (Table A1, Table A2, Table A3 and Table A4). Axial stiffness and cycles to failure demonstrated normal data distribution, whereas displacements at the fracture sites were not normally distributed.

### 3.1. Axial Stiffness

Greatest initial axial stiffness was observed in the S1/S2 ala–ilium group (18.76 ± 5.77 N/mm), compared to all other groups (*p* < 0.01) (Table 1, Figure 5). Axial stiffness in the transpubic screw group (9.10 ± 2.44 N/mm) was significantly greater compared to the SI screw group (3.79 ± 1.60 N/mm) (*p* < 0.01). Axial stiffness in the plate group (8.18 2.81 ± N/mm) was greater compared to the SI screw group 1 (3.79 ± 1.60 N/mm); however, this difference was not significant (*p* = 0.06). No further significant differences regarding axial stiffness were detected (*p* > 0.71).

### 3.2. Cycles to Failure

Cycles to failure were greatest in the plate group (7647 cycles) and in the transpubic screw group (8237 cycles) compared to all other groups (*p* < 0.01) (Table 1, Figure 6). Furthermore, cycles to failure were significantly higher in the S1/S2 ala–ilium group (3863 cycles) compared to the SI screw group (856 cycles) (*p* < 0.01) and compared to the external fixator group (1879) (*p* = 0.03). No significant differences regarding cycles to failure were detected between the SI screw group (856 cycles) and the external fixator (1879 cycles) groups (*p* > 0.999).

### 3.3. Fracture Displacement

The plate group, the transpubic screw group, and the S1/S2 ala–ilium group had significantly less displacement at the anterior fracture sites compared to the SI screw group and the external fixator group (*p* ≤ 0.01) (Table 2).

Regarding the displacement at the posterior fracture site, the plate group and the transpubic screw group had significantly less displacement compared to the SI screw group (*p* ≤ 0.04). Furthermore, the S1/S2 ala–ilium group had significantly less displacement at the posterior fracture site compared to the SI screw group and the external fixator group (*p* ≤ 0.01).

The failure mode for all specimens of the SI screw group and the external fixator group was similar (Figure 7), as follows: the machine transducer reached 30 mm displacement with failure of the anterior and posterior pelvic ring. The failure mode for all specimens of the plate group and the transpubic screw group was also comparable (Figure 7), as follows: the machine transducer reached 30 mm displacement with failure of the posterior pelvic ring. The failure mode for all specimens of the S1/S2 ala–ilium group was similar, with failure of the anterior and posterior pelvic ring and the machine transducer reaching 30 mm displacement. No implant breakage or screw loosening were observed.

## 4. Discussion

In the current study, the techniques using SI screw plus plate fixation and SI screw plus retrograde transpubic screw achieved greatest overall stability. Thus, the hypothesis that supplemental plate fixation of the anterior pelvic ring would be superior to both the supplemental external fixation and the standalone S1/S2 ala–ilium screws was confirmed. However, no significant differences were found between the SI screw plus plate fixation and SI screw plus retrograde transpubic screw techniques. The S1/S2 ala–ilium screws achieved the greatest initial axial stiffness of all of the tested techniques, and were superior both to the SI screw alone and to the SI screw plus supra-acetabular external fixator.

In general, non-operative treatment of FFPs is considered to be associated with fewer complications than operative treatment [18], though it is reported that the two-year survival rate is significantly higher in older patients with surgically fixated pelvic fractures [19]. The choice of treatment for an FFP depends on the possibility of mobilising the patient [8]. Re-establishing mobility with as little additionally morbidity as possible, as well as shorter duration of the surgical procedures, are both crucially important [8]. It has been shown that the percutaneous SI screw insertion is a safe and effective surgical procedure for stabilisation of the posterior pelvic ring [20]. However, there is insufficient evidence in the literature for the superiority of one the literature for the superiority [8], and it remains unclear whether simultaneous stabilisation of the anterior pelvic ring should be performed in anterior and posterior pelvic ring injuries [9], although this is recommended from a clinical point of view [10].

The present study demonstrated that the simultaneous stabilisation of the anterior and posterior pelvic ring increased the overall pelvic ring stability, and that anterior plating or the use of a retrograde transpubic screw led to substantially less displacement of the anterior pelvic ring, and therefore to higher stability compared with all of the other tested techniques. These findings are consistent with the results of previous biomechanical studies. In human cadaveric type C pelvis fractures, internal fixation (sacral bars, SI plates, screws, etc.) in addition to external fixation led to significantly greater stability [21]. For cadaveric sacral fractures, Stocks et al. [22] demonstrated that anterior plating in combination with posterior rods was significantly more stable than sacral rods with an anterior Hoffmann frame. Further evidence showing the biomechanical importance of simultaneous stabilisation is brought by the fact that the S1/S2 ala–ilium screws lost more overall fixation strength over time than both of the methods with concomitant internal fixation of the anterior pelvic ring. This result is in agreement with a previous biomechanical study showing the significant increase of overall stability when an anterior plate is used simultaneously with a lumbopelvic fixation [23].

No significant differences between anterior plating and retrograde transpubic screw were found in the present study. This result is consistent with the biomechanical results of the previous study of Simonian et al. [24]. In cadaveric specimens, plating and retrograde transpubic screws were biomechanically comparable in addressing simulated APC-II pelvic ring injuries. In a further biomechanical study, using synthetic pelvic bone models and simulating type C pelvic ring fracture, the fixation with retrograde transpubic screw was also biomechanically equivalent to the plate fixation [14].

The greatest initial axial stiffness in the present study was achieved using S1/S2 ala–ilium screws. Previous biomechanical studies testing lumbosacral fixation methods reported comparable results. Cunningham et al. [25] tested different fixation methods on porcine lumbosacral spines, showing that the iliac screw construct was the most restrictive with respect to the motion at the lumbosacral joint. Jazini et al. [23] performed a biomechanical study on the role of lumbopelvic fixation in sacral fractures in cadaveric specimens. Lumbopelvic fixation in combination with a transiliac–transsacral screw resulted in the least amount of motion [23]. However, the used S1/S2 ala–ilium screws technique in this study was restricted to sacral fixation, without including the lumbar spine, as described elsewhere [12]. The S1/S2 ala–ilium screws were significantly superior in terms of cycles to failure, corresponding failure load, and initial axial stiffness compared to the posterior SI screw alone and to the SI screw plus supra-acetabular external fixator. In displaced unstable pelvic ring injuries, external fixation fails to provide sufficient posterior compressive force [26,27]. Gardner et al. [28] demonstrated in their biomechanical study simulating a symphyseal and unilateral sacroiliac joint disruption in synthetic pelvises that a standard two-bar external fixation did not apply any compression across the sacroiliac joint. In a clinical study, the minimally invasive S2 ala–ilium screw fixation has been described as a successful treatment option for pelvic trauma [12]. This technique might be a successful treatment option for FFP type IIc. However, the treatment of geriatric pelvic ring fractures with a percutaneous SI screw and external fixator for the anterior ring is clinically recommended [10,19,29].

From a clinical point of view, some authors prefer less invasive techniques when treating elderly patients with FFPs, including using transpubic screws [8]. Surgical treatment complications are mainly approach-related, and occur in up to 30% of cases [10]. Therefore, minimising the surgical approach reduces the rate of complications [8,10]. In the presence of dislocated anterior pelvic ring fractures, open approaches and reduction are necessary. However, anatomical reduction in the elderly is considered to be unnecessary, and manipulation with a short screw can be performed [30,31]. Furthermore, most superior pubic ramus fractures can be treated minimally invasively using a retrograde transpubic screw [32]. Our results demonstrating the biomechanical equality of plating and retrograde screw fixation may contribute to this discussion, and hopefully facilitate clinical decision making.

The most notable limitation of this biomechanical study is the use of synthetic bones without soft tissue, muscles, and in particular ligaments, which may have resulted in outcomes differing from those compared to human cadaveric pelvises [33,34,35]. Simulating an FFP type II via an osteotomy of the anterior and posterior pelvic ring as described above most closely corresponds to an FFP type IIc. However, the fractures set via a saw cut are different from fractures resulting from low-energy trauma. Pre-testing showed the necessity of bonding the symphysis in order to increase the stability of the models [14]. The curved rod of the external fixator [36] was necessary in order to ensure compatibility with motion tracking. Fracture displacements were measured after 500 cycles only—being too big for capturing by the system cameras after higher number of cycles in groups 1 and 2. However, synthetic bones have less inter-specimen variability compared with their organic counterparts, thus reducing the heterogeneity in size and bone quality [28]. Another strength of this study is the large number of tested specimens (Figure A8). Furthermore, a high reliability of the conducted procedures was achieved using standardised methods, such as individually customised PMMA guides for osteotomy setting and implantation (Figure A1, Figure A2, Figure A3, Figure A4, Figure A5, Figure A6, Figure A7 and Figure A8).

However, a further biomechanical study using human cadaveric specimens is needed in order to confirm the results of the present study. Furthermore, clinical studies comparing the examined techniques might validate the results of the present study.

All in all, the present biomechanical study showed that the minimally invasive techniques were as stiff and stable during dynamic testing as the more invasive plating technique, and could help find optimal choice for the treatment of FFP.

## 5. Conclusions

In the present model, the simultaneous fixation of the anterior and posterior pelvic ring simultaneously provided higher overall stability. The use of external fixator increased the overall stability to some extent. The SI screw plus plate fixation and the SI screw plus retrograde transpubic screw led to higher stability than the other tested techniques. The S1/S2 ala–ilium screws were more stable than both the SI screw alone and the SI screw plus external fixator. The results of the present study could help find the optimal choice for treatment of FFP type II, showing excellent biomechanical competence of the minimally invasive techniques. In cases with displaced fractures, open reduction and fixation with a plate provides the highest stability, whereas in cases where minimally invasive techniques are applicable, the retrograde transpubic screw and the S1/S2 ala–ilium screws can be considered as successful alternative treatment options.

## Figures and Tables

**Figure 1 jcm-10-02326-f001:**
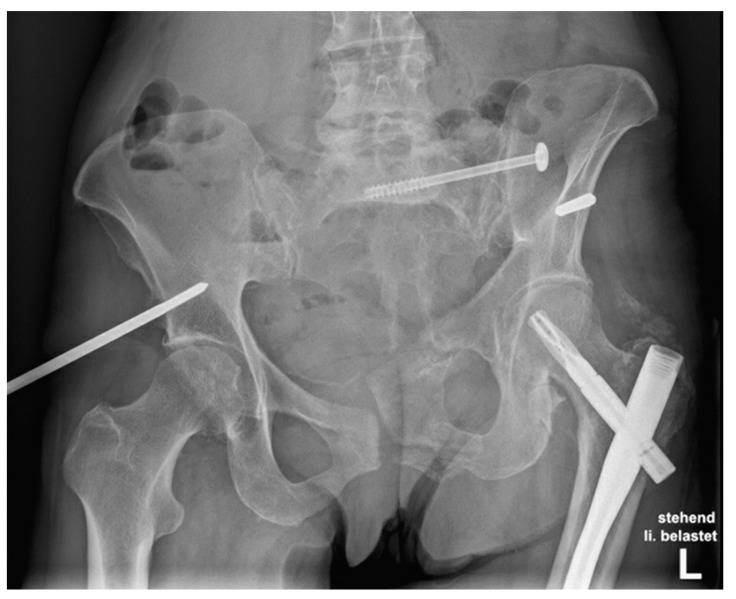
Fixation failure in a 66-year-old patient 3 months after osteosynthesis of a pelvic ring fracture with a vertical fracture of the sacrum ((FFP) type IIc). The anterior ring was addressed with an external fixator, whereas the posterior pelvic ring was addressed with an SI screw.

**Figure 2 jcm-10-02326-f002:**
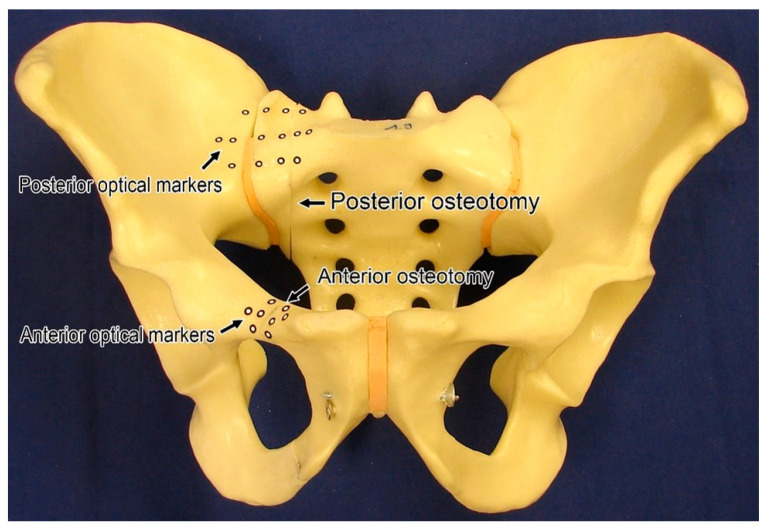
Anterior view of an artificial pelvis with anterior and posterior osteotomies simulating type II fragility fracture of the pelvis, and optical markers for motion tracking during biomechanical testing. The pubic symphysis and the contralateral (**left**) sacroiliac joint are bonded with coarse-threaded screws for specimen standardization and stiffening.

**Figure 3 jcm-10-02326-f003:**
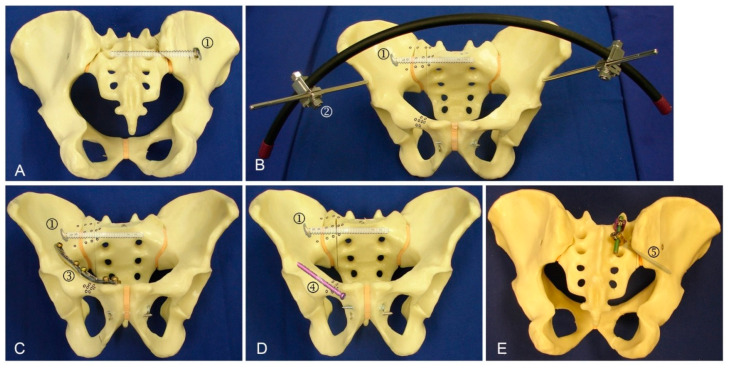
Exemplified specimens from the 5 study groups instrumented with (**A**) group 1: SI screw (1); (**B**) group 2: SI screw (1) plus supra-acetabular external fixator (2); (**C**) group 3: SI screw (1) plus plate fixation (3); (**D**) group 4: SI screw (1) plus retrograde transpubic screw (4); (**E**) group 5: S1/S2 ala–ilium screws (5).

**Figure 4 jcm-10-02326-f004:**
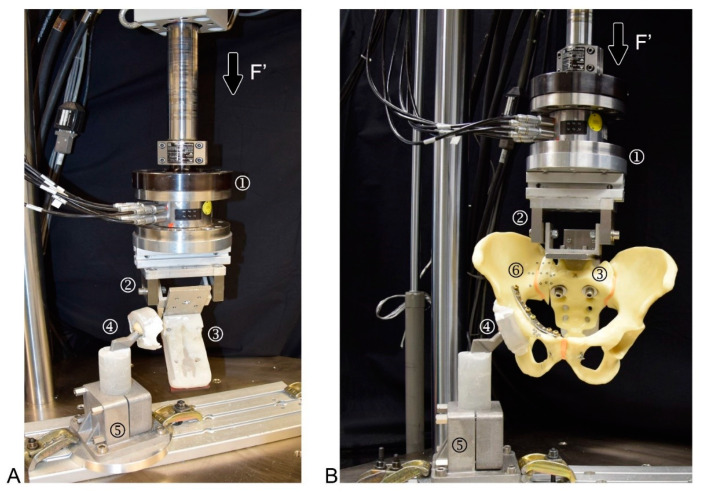
(**A**): Setup for biomechanical testing in a one-legged stance position of the specimen (with no speciment mounted). Force (F’) was introduced by axial compression and recorded by a load cell (1) connected to a hinge (2) for rigid connection to the specimen using a customized PMMA cast for sacrum fixation (3). One-legged stance position was simulated using a hemiarthroplasty (4) rigidly mounted on the machine base (5) (**B**): Identical setup to that visualised in A, with a mounted specimen, including optical tracking markers (6).

**Figure 5 jcm-10-02326-f005:**
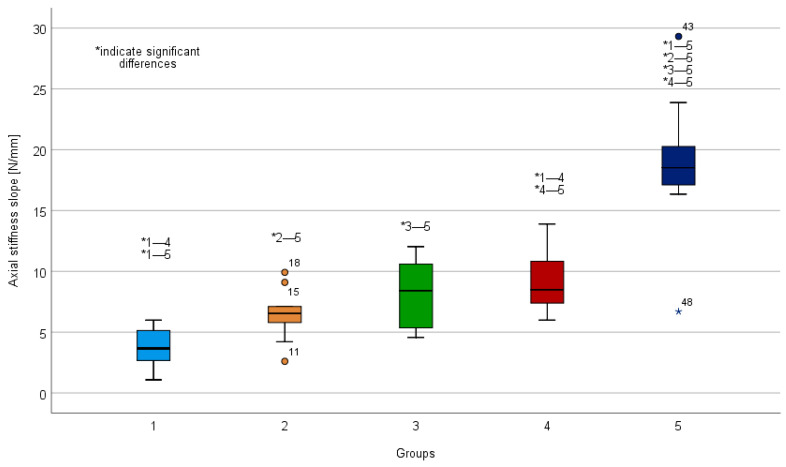
Boxplots showing initial axial stiffness slope. 1: Sacroiliac (SI) screw; 2: SI screw plus supra-acetabular external fixator; 3: SI screw plus anterior plate; 4: SI screw plus retrograde transpubic screw; 5: S1/S2 ala–ilium screws. Asterisks above the boxplots indicate significant differences.

**Figure 6 jcm-10-02326-f006:**
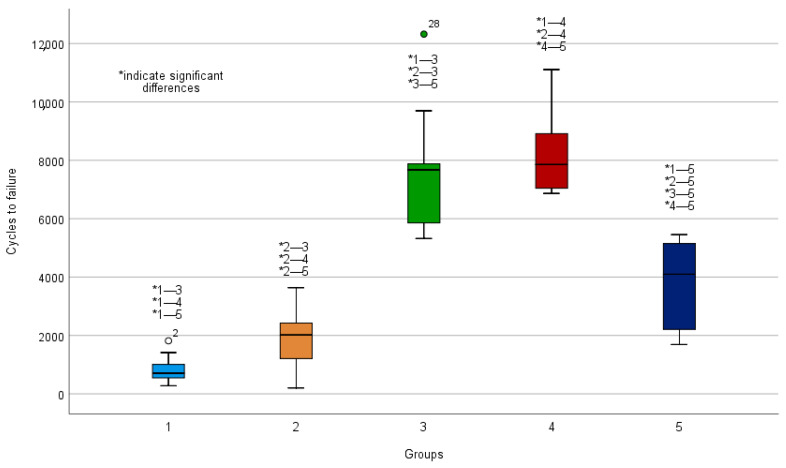
Boxplots showing cycles to failure. 1: Sacroiliac (SI) screw; 2: SI screw plus supra-acetabular external fixator; 3: SI screw plus anterior plate; 4: SI screw plus retrograde transpubic screw; 5: S1/S2 ala–ilium screws. Asterisks above the boxplots indicate significant differences.

**Figure 7 jcm-10-02326-f007:**
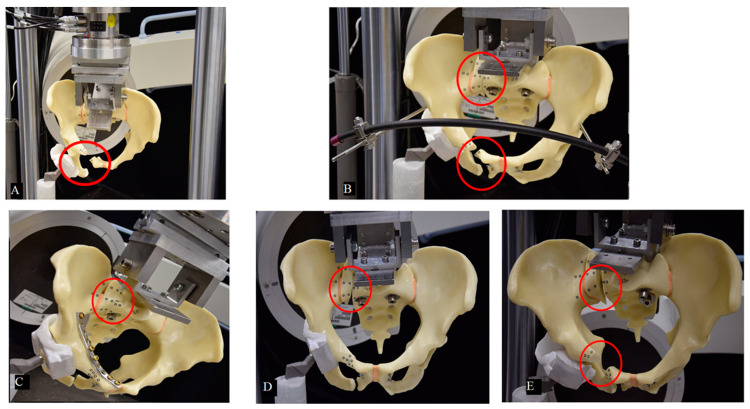
Failure modes (red circle) in groups 1 (SI screw) (**A**), 2 (SI screw plus supra-acetabular external fixator) (**B**), 3 (SI screw plus plate fixation) (**C**), 4 (SI screw plus a retrograde transpubic screw) (**D**), and 5 (S1/S2 ala–ilium screws) (**E**) after reaching test stop criterion of 30 mm machine actuator displacement.

**Table 1 jcm-10-02326-t001:** Outcome measures for axial stiffness and cycles to failure.

Groups 1–5	Initial Axial Stiffness [N/mm]	Cycles to Failure
SI screw (1); mean (SD)	3.79 (1.60)	856 (470)
SI screw plus supra-acetabular external fixator (2); mean (SD)	6.52 (2.11)	1879 (1027)
SI screw plus plate fixation (3); mean (SD)	8.18 (2.81)	7647 (2248)
SI screw plus a retrograde transpubic screw (4); mean (SD)	9.10 (2.44)	8237 (1399)
S1/S2 ala–ilium screw fixation (5); mean (SD)	18.76 (5.77)	3863 (1476)

**Table 2 jcm-10-02326-t002:** Outcome measures for displacement at the fracture sites, shown in terms of median and interquartile range (IQR).

Groups 1–5	Anterior Displacement at 500 Cycles [cm]	Posterior Displacement at 500 Cycles [cm]
SI screw (1); median (IQR)	5.74 (1.43–8.14)	0.41 (0.16–0.67)
SI screw plus supra-acetabular external fixator (2); median (IQR)	4.33 (2.42–6.27)	0.30 (0.16–0.43)
SI screw plus plate fixation (3); median (IQR)	0.01 (0.00–0.02)	0.00 (0.00–0.11)
SI screw plus a retrograde transpubic screw (4); median (IQR)	0.01 (0.00–0.01)	0.00 (0.00–0.04)
S1/S2 ala–ilium screw fixation (5); median (IQR)	0.00 (0.00–0.15)	0.00 (0.00–0.01)

## Data Availability

Data are contained within the article.

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
