# Peer review of "Biomechanical Comparison of Five Fixation Techniques for Unstable Fragility Fractures of the Pelvic Ring"

_jcm, 2021, doi:10.3390/jcm10112326_

Round 1

Reviewer 1 Report

A fragility fracture of the pelvis  (FFP) type IIc ( non-displaced sacral, iliosacral or ilium fracture with anterior disruption)  was simulated in 50 artificial pelvises. These fractures  were assigned to 5 study groups: Sacroiliac- (SI) screw, SI-screw plus supra-acetabular external fixator, SI-screw plus plate, SI-screw plus retrograde transpubic screw, or S1/S2-ala-  ilium screws. The specimens were tested under progressively increasing cyclic loading.  The  cycles to failure and axial stiffness were analysed.  In this biomechanical  study the SI-screw plus plate fixation and the SI-screw plus retrograde transpubic screw showed  higher stability than the other tested techniques. The S1/S2-ala-ilium screws were  more stable than the SI-screw or the SI-screw plus external fixator.  And  the  authors  concluded that in cases  of displaced fractures ORIF  with a plate provides highest stability  whereas in cases where minimally invasive techniques can be used the retrograde  transpubic screw and the S1/S2-ala-ilium screws can be considered as successful alternative treatment options.

This  is  an interesting  investigation particularly because  similar  studies  in osteoporotic  saw  bane pelvis  are  sparse.  The  use of  an FFP type II was simulated at the right side of 50 artificial pelvises (osteoporotic  bone Model)  is  the  correct  method for such  biomechanical studies  to reduce  variability  linked  with  cadaveric  pelvis. Surely  the  osteotomies  made  by a  chisel may be  not  quite  similar  to the  fractures  acquired   during  a fall, however  sufficient  for  this  investigation.

I  am wondering  why the  spinopelvic  method of  fixation was not  also  tested ?

An  another  question  is  how  to reduce  these  displaced  fractures  in osteoporotic  pelvis?

I  have  some  concern  regarding  the  fixation model No 4 (S1/S2-  ala-ilium screws (group 5, S1/S2-ala-ilum group).  Very often this  is not possible  because of  the  severely osteoporotic  bone  affecting  sacrum.

Generally, the  results  were  the  anticipated  for  unstable  pelvic  ring  fractures  even in elderly.

Author Response

Response to Reviewer 1 Comments

Dear Editor,

thank you for your kind letter concerning our manuscript. We were happy to see the encouraging comments and wise instructions by the reviewers. Below, we comment every question and suggestion and show the changes we have made in the revised manuscript that is enclosed.

Point 1: This is an interesting investigation particularly because similar studies in osteoporotic saw bane pelvis are sparse. The use of an FFP type II was simulated at the right side of 50 artificial pelvises (osteoporotic bone Model) is the correct method for such biomechanical studies to reduce variability linked with cadaveric pelvis. Surely the osteotomies made by a chisel may be not quite similar to the fractures acquired during a fall, however sufficient for this investigation.

We would like to thank reviewer 1 for the positive and constructive feedback and discuss this point in the limitation section. Please see ll. 339-342.

Point 2: I am wondering why the spinopelvic method of fixation was not also tested?

We appreciate the reviewer for bringing this up. On the one hand it was shown in a clinical study that a solely pelvic fixation method represents a successful surgical technique for isolated pelvic ring injuries. (Hartensuer et al. 2020). On the other hand, the present specific biomechanical model was limited to the pelvis without the lumbar spine to evaluate common pelvic fixation methods. However, this point is of great interest and in a further biomechanical project the spinopelvic fixation method should be compared to the techniques tested in the present study.

Point 3: Another question is how to reduce these displaced fractures in osteoporotic pelvis?

Thank you for highlighting this. The reduction of displaced fractures especially in osteoporotic pelvises is difficult. Please see our discussion ll. 329-336:

“In the presence of dislocated anterior pelvic ring fractures, open approaches and reduction is necessary. However, anatomical reduction in the elderly is considered to be not necessary and in difficult cases retro-ante-retrograde technique or manipulation with a short screw can be performed (Weatherby et al. 2017; Mosheiff und Liebergall 2002). Furthermore, most superior pubic ramus fractures can be treated minimally invasive using a retrograde transpubic screw (Rommens et al. 2020). Our results of biomechanical equality of plating and retrograde screw fixation may contribute to this discussion and hopefully facilitate clinical decision making.”

We prefer the open reduction in displaced pelvis fractures as we highlight this in our manuscript, please see ll. 365-369. From our point of view reduction of osteoporotic pelvis should be performed with similar techniques as reduction of non-osteoporotic pelvis. However, utmost care is mandatory during reduction not to further displace fracture fragments, nor worsen the comminution.

Point 4: I have some concern regarding the fixation model No 4 (S1/S2- ala-ilium screws (group 5, S1/S2-ala-ilum group). Very often this is not possible because of the severely osteoporotic bone affecting sacrum.

Thank you for raising this point. It was described in literature that especially the S1/S2-ala-ilium screw represents a valuable alternative in cases of osteoporosis and for revision surgery for fracture non-union of the posterior pelvic ring after previous percutaneous SI screw fixation (Hartensuer et al. 2020). Hereby, the main anchorage of the screw is provided by passing the Si-Joint with its two cortical walls and by getting trapped in the iliac bone, using 8 to 10mm diameter screws. This seems also effective also in osteoporosis. The S1 pedicel screws need to be placed bi-cortical. In our experience the sacral starting point is not mandatory for stability but for optimized implant placement with less soft tissue affection. It therefore provides an alternative, or an addition, to previous management strategies for treating injuries of the posterior pelvic ring and the spinopelvic junction. However, larger clinical studies underlining the successful use for this technique for treatment of pelvic ring injuries are necessary.

Point 5: Generally, the results were the anticipated for unstable pelvic ring fractures even in elderly.

We thank the reviewer for this suggestion. However, it is described that there is a lack of evidence for the superiority of one technique over another (Rommens et al. 2017). Furthermore, to the best of our knowledge it was not shown before that anterior plating and the retrograde transpubic screw were comparable for treatment of FFP II as well as that the S2-ala-ilium screw was biomechanically superior compared to the external fixator and the SI-screw. We believe that this manuscript can contribute substantially to the evolution of treatment of FFP.

Literaturverzeichnis

Hartensuer, Rene; Grüneweller, Niklas; Lodde, Moritz Friedrich; Evers, Julia; Riesenbeck, Oliver; Raschke, Michael (2020): The S2-Alar- Iliac Screw for Pelvic Trauma. In: Zeitschrift fur Orthopadie und Unfallchirurgie. DOI: 10.1055/a-1190-5987.

Mosheiff, Rami; Liebergall, Meir (2002): Maneuvering the retrograde medullary screw in pubic ramus fractures. In: Journal of orthopaedic trauma 16 (8), S. 594–596. DOI: 10.1097/00005131-200209000-00009.

Rommens, Pol M.; Graafen, Marcus; Arand, Charlotte; Mehling, Isabella; Hofmann, Alexander; Wagner, Daniel (2020): Minimal-invasive stabilization of anterior pelvic ring fractures with retrograde transpubic screws. In: Injury 51 (2), S. 340–346. DOI: 10.1016/j.injury.2019.12.018.

Rommens, Pol Maria; Wagner, Daniel; Hofmann, Alex (2017): Minimal Invasive Surgical Treatment of Fragility Fractures of the Pelvis. In: Chirurgia (Bucharest, Romania : 1990) 112 (5), S. 524–537. DOI: 10.21614/chirurgia.112.5.524.

Weatherby, David J.; Chip Routt, Milton L.; Eastman, Jonathan G. (2017): The Retrograde-Antegrade-Retrograde Technique for Successful Placement of a Retrograde Superior Ramus Screw. In: Journal of orthopaedic trauma 31 (7), e224-e229. DOI: 10.1097/BOT.0000000000000849.

Reviewer 2 Report

I congratulate the authors for a well executed and presented biomechanical study. 

The results could be presented slightly more clearly; in the figures, an asterisk above the boxplot with the corresponding numbers in parentheses would indicate the signifant comparisons more clearly than as currently reflected.

For tables 1 and 2, it is customary to include the mean / median and deviation / IQR in a single line, with the latter two in parenteses. This makes comparisons in the same column easier. 

Tables in the Addendum do not have units for the values, and the table legends should be expressed more clearly. 

Finally, many FFP II fractures of the pelvic ring have incomplete sacral fractures (compression mechanism). I understand the experimental design included a complete osteotomy of the sacral wing. This should be mentioned as a limitation. 

Author Response

Response to Reviewer 2 Comments

Dear Editor,

thank you for your kind letter concerning our manuscript. We were happy to see the encouraging comments and wise instructions by the reviewers. Below, we comment every question and suggestion and show the changes we have made in the revised manuscript that is enclosed.

Point 1: The results could be presented slightly more clearly; in the figures, an asterisk above the boxplot with the corresponding numbers in parentheses would indicate the significant comparisons more clearly than as currently reflected.

We would like to thank reviewer 2 for the positive and constructive feedback. We added an asterisk above the boxplot with the corresponding numbers in parentheses indicating the significant comparisons.

Point 2: For tables 1 and 2, it is customary to include the mean / median and deviation / IQR in a single line, with the latter two in parentheses. This makes comparisons in the same column easier.

Thank you for raising this point. The mean / median and deviation / IQR are listed in one line with the latter two in parentheses.

Point 3: Tables in the Addendum do not have units for the values, and the table legends should be expressed more clearly.

We appreciate the reviewer for bringing this up and we added the units for the values as well as we changed the table legends.

Point 4: Finally, many FFP II fractures of the pelvic ring have incomplete sacral fractures (compression mechanism). I understand the experimental design included a complete osteotomy of the sacral wing. This should be mentioned as a limitation.

Thank you for highlighting this and we agree with Reviewer 2 and we added this in the limitation section. Please see ll. 337-340.
